# CIGMO: Learning categorical invariant deep generative models from grouped data

## Abstract

Images of general objects are often composed of three hidden factors: category (e.g., car or chair), shape (e.g., particular car form), and view (e.g., 3D orientation). While there have been many disentangling models that can discover either a category or shape factor separately from a view factor, such models typically cannot capture the structure of general objects that the diversity of shapes is much larger across categories than within a category. Here, we propose a novel generative model called CIGMO, which can learn to represent the category, shape, and view factors at once only with weak supervision. Concretely, we develop mixture of disentangling deep generative models, where the mixture components correspond to object categories and each component model represents shape and view in a category-specific and mutually invariant manner. We devise a learning method based on variational autoencoders that does not explicitly use label information but uses only grouping information that links together different views of the same object. Using several datasets of 3D objects including ShapeNet, we demonstrate that our model often outperforms previous relevant models including state-of-the-art methods in invariant clustering and one-shot classification tasks, in a manner exposing the importance of categorical invariant representation.

## 1 Introduction

In everyday life, we see objects in a great variety. Categories of objects are numerous and their shape variations are tremendously rich; different views make an object look totally different (Figure 1(A)). Recent neuroscientific studies have revealed how the primate brain organizes representation of complex objects in the higher visual cortex (Freiwald & Tsao, 2010; Srihasam et al., 2014; Bao et al., 2020). According to these, it comprises multi-stream networks, each of which is specialized to a particular object category, encodes category-specific visual features, and undergoes multiple stages with increasing view invariance. These biological findings inspire us a new form of learning model that has multiple modules of category-specific invariant feature representations.

More specifically, our goal is, given an image dataset of general objects, to learn a generative model representing three latent factors: (1) category (e.g., cars, chairs), (2) shape (e.g., particular car or chair types), and (3) view (e.g., 3D orientation). A similar problem has been addressed by recent disentangling models that discover complex factors of input variations in a way invariant to each other (Tenenbaum & Freeman, 2000; Kingma et al., 2014; Chen et al., 2016; Higgins et al., 2016; Bouchacourt et al., 2018; Hosoya, 2019). However, although these models can effectively infer a category or shape factor separately from a view factor, these typically cannot capture the structure in general object images that the diversity of shapes is much larger across categories than within a category.

In this study, we propose a novel model called CIGMO (Categorical Invariant Generative MOdel), which can learn to represent all the three factors (category, shape, and view) at once only with weak supervision. Our model has the form of mixture of deep generative models, where the mixture components correspond to categories and each component model gives a disentangled representation of shape and view for a particular category. We develop a learning algorithm based on variational autoencoders (VAE) method (Kingma & Welling, 2014) that does not use explicit labels, but uses only grouping information that links together different views of the same object (Bouchacourt et al., 2018; Chen et al., 2018; Hosoya, 2019).

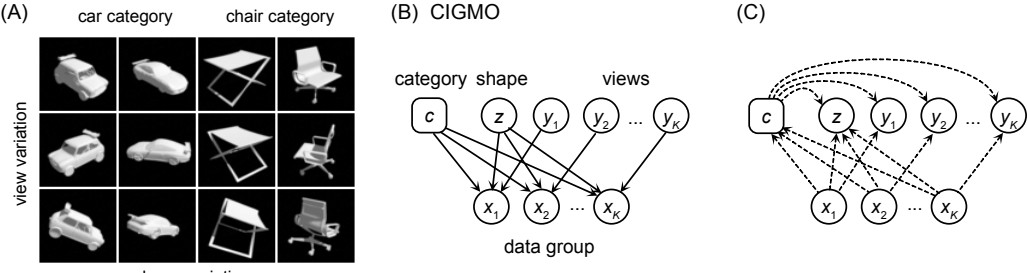

Figure 1: (A) Examples of general object images. These include two categories (car and chair) each with two shape variations. In addition, the object of each shape is shown in three different views. (B) The graphical model. Each instance $x_k$ in a data group is generated from a category $c$, a shape $z$, and a view $y_k$. Round boxes are discrete variables and circles are continuous variables. (C) The inference flow. Each hidden variable is inferred from the set of incoming variables.

Using two image datasets of 3D objects (one derived from ShapeNet (Chang et al., 2015)), we demonstrate that CIGMO can solve multiple unconventional visual tasks on objects that are unseen during training. These include invariant clustering, i.e., clustering of objects regardless of the view, one-shot classification, i.e., object recognition given one example per class, and other various feature manipulations using the disentangled representation. Quantitative comparison indicates that our model often outperforms many existing approaches including state-of-the-art methods.

Our key contributions are (1) development of the new deep generative model CIGMO, together with the VAE-based weakly supervised learning algorithm, (2) experiments of the model on two datasets with quantitative comparisons that show performance advantages over several existing models, and (3) empirical exposition of the importance of separate representation of category, shape, and view factors in modeling general object images.

## 2 RELATED WORK

The present work is closely related to recently proposed disentangling models for discovering mutually invariant factors of variation in the input. In one direction, some models have used unsupervised learning with certain constraints on the latent variable, though these seem to be effective only in limited cases (Higgins et al., 2016; Chen et al., 2016). Thus, more practical approaches have made use of explicit labels, such as semi-supervised learning for a part of dataset (Kingma et al., 2014; Siddharth et al., 2017) or adversarial learning to promote disentanglement (Lample et al., 2017; Mathieu et al., 2016). However, labels are often expensive.

To find out a good compromise, weaker forms of supervision have been investigated. One such direction is group-based learning, which assumes inputs with the same shape to be grouped together (Bouchacourt et al., 2018; Chen et al., 2018; Hosoya, 2019). In particular, our study here is technically much influenced by Group-based VAE (GVAE) (Hosoya, 2019) in the algorithm construction (Section 3). However, these existing group-based methods are fundamentally limited in that the factors that can be separated are two—a group-common factor (shape) and an instance-specific factor (view)—and there is no obvious way to extend it to more than two. Thus, our novelty here is to introduce a mixture model comprising multiple GVAE models (each with shape and view variables) so that fitting the mixture model to a grouped dataset can give rise to the third factor, categories, as mixture components. In Section 4, we examine the empirical merits of this technique in several tasks. Note that grouping information can most naturally be found in temporal data (like videos) since the object identity is often stable over time, cf. classical temporal coherence principle (Földiák, 1991). Indeed, some weakly supervised disentangling approaches have explicitly used such temporal structure (Yang et al., 2015).

Some recent work has used deep nets for clustering of complex object images. The most typical approach is a simple combination of a deep generative model (e.g., VAE) and a conventional clustering method (e.g., Gaussian mixture), although such approach seems to be limited in capturing

large object view variation (Jiang et al., 2017). A latest approach proposes a feedforward approach that takes pairs of image data, similarly to ours, and maximizes the mutual information between the categorical posterior probability distributions for such paired image; this has shown remarkable clustering performance on natural images under various view variation (Ji et al., 2019). In Section 4, we experimentally compare their method with ours. Note, however, that these methods are specialized to clustering and throw away all information other than the category.

## 3 CIGMO: CATEGORICAL INVARIANT GENERATIVE MODEL

### 3.1 MODEL

In our framework, we assume a grouped dataset

$$\mathbb{D} = \{(\boldsymbol{x}_1^{(n)}, \ldots, \boldsymbol{x}_K^{(n)}) \mid \boldsymbol{x}_k^{(n)} \in \mathbb{R}^D, n = 1, \ldots, N\} \tag{1}$$

where each data point is a group (tuple) of $K$ data instances (e.g., images); we assume independence between groups but not instances within a group. For a data group $(\boldsymbol{x}_1, \ldots, \boldsymbol{x}_K)$, we consider three types of hidden variables: category $c \in \{1, \ldots, C\}$, shape $\boldsymbol{z} \in \mathbb{R}^M$, and views $\boldsymbol{y}_1, \ldots, \boldsymbol{y}_K \in \mathbb{R}^L$ (eliding the superscript $(n)$ for brevity), where the category and shape are common for the group while the views are specific to each instance. We consider the following generative model (Figure 1(B)):

$$p(c) = \pi_c \tag{2}$$
$$p(\boldsymbol{z}) = \mathcal{N}_M(0, \boldsymbol{I}) \tag{3}$$
$$p(\boldsymbol{y}_k) = \mathcal{N}_L(0, \boldsymbol{I}) \tag{4}$$
$$p(\boldsymbol{x}_k|\boldsymbol{y}_k, \boldsymbol{z}, c) = \mathcal{N}_D(f_c(\boldsymbol{y}_k, \boldsymbol{z}), \boldsymbol{I}) \tag{5}$$

for $c = 1, \ldots, C$ and $k = 1, \ldots, K$. Here, $f_c$ is a decoder deep net defined for each category $c$ and $\pi_c$ is a category prior with $\sum_{c=1}^{C} \pi_c = 1$. In the generative process, first, the category $c$ is drawn from the categorical distribution $(\pi_1, \ldots, \pi_C)$, while the shape $\boldsymbol{z}$ and views $\boldsymbol{y}_k$ are drawn from standard Gaussian priors. Then, each data instance $\boldsymbol{x}_k$ is generated by the decoder deep net $f_c$ for category $c$ applied to the group-common shape $\boldsymbol{z}$ and the instance-specific view $\boldsymbol{y}_k$ (added with Gaussian noise of unit variance). In other words, the decoder $f_c$ generates different data instances in a group from the same shape and different views. Having defined a mixture of deep generative models as above, we expect that, after fitting it to a view-grouped object image dataset, object categories will arise as mixture components and category-specific shapes and views will be represented in each component model.

### 3.2 LEARNING

We construct a learning algorithm following the VAE approach (Kingma & Welling, 2014). As the most important step, we specify inference models to encode approximate posterior distributions (Figure 1(C)). First, we estimate the posterior probability for category $c$ as follows:

$$q(c|\boldsymbol{x}_1, \ldots, \boldsymbol{x}_K) = \frac{1}{K} \sum_{k=1}^{K} u^{(c)}(\boldsymbol{x}_k) \tag{6}$$

Here, $u$ is a classifier deep net that takes an individual instance $\boldsymbol{x}$ and outputs a probability distribution over the categories ($\sum_{c=1}^{C} u^{(c)}(\boldsymbol{x}) = 1$), similarly to (Kingma et al., 2014). We then take the average over the instance-specific probability distributions and use it as the group-common distribution. This is a simple approach to make the instance-specific distributions converge to equal values, i.e., $u(\boldsymbol{x}_1) \approx u(\boldsymbol{x}_2)$. This is an adaptation of a key technique of GVAE used for computing the group-common shape representation (Hosoya, 2019); see below.

Then, given the estimated category $c$, we infer each instance-specific view $\boldsymbol{y}_k$ from the input $\boldsymbol{x}_k$ as follows:

$$q(\boldsymbol{y}_k|\boldsymbol{x}_k, c) = \mathcal{N}_L\left(g_c(\boldsymbol{x}_k), \mathrm{diag}(r_c(\boldsymbol{x}_k))\right) \tag{7}$$

where $g_c$ and $r_c$ are encoder deep nets that are defined for each category $c$ to specify the mean and variance, respectively. To estimate shape $\boldsymbol{z}$, we compute the following:

$$q(\boldsymbol{z}|\boldsymbol{x}_1,\ldots,\boldsymbol{x}_K,c) = \mathcal{N}_M\left(\frac{1}{K}\sum_{k=1}^{K}h_c(\boldsymbol{x}_k), \frac{1}{K}\sum_{k=1}^{K}\text{diag}(s_c(\boldsymbol{x}_k))\right) \tag{8}$$

Here, again, encoder deep nets $h_c$ and $s_c$ are defined for each category $c$. These compute the mean and variance, respectively, of the posterior distribution for the individual shape for each instance $\boldsymbol{x}_k$. Then, the group-common shape $\boldsymbol{z}$ is obtained as the average over all the individual shapes. In this way, again, the instance-specific shape representations are expected to converge to an equal value in the course of training, i.e., $h_c(\boldsymbol{x}_1) \approx h_c(\boldsymbol{x}_2)$ (Hosoya, 2019). Note that the way the view and shape are inferred here is mostly borrowed from Hosoya (2019).

For training, we define the following variational lower bound of the marginal log likelihood for a data point:

$$\mathcal{L}(\phi;\boldsymbol{x}_1,\ldots,\boldsymbol{x}_K) = \mathcal{L}_{\text{recon}} + \mathcal{L}_{\text{KL}} \tag{9}$$

where

$$\mathcal{L}_{\text{recon}} = \mathbb{E}_{q(\boldsymbol{y}_1,\ldots,\boldsymbol{y}_K,\boldsymbol{z},c|\boldsymbol{x}_1,\ldots,\boldsymbol{x}_K)}\left[\sum_{k=1}^{K}\log p(\boldsymbol{x}_k|\boldsymbol{y}_k,\boldsymbol{z},c)\right] \tag{10}$$

$$\mathcal{L}_{\text{KL}} = -D_{\text{KL}}(q(\boldsymbol{y}_1,\ldots,\boldsymbol{y}_K,\boldsymbol{z},c|\boldsymbol{x}_1,\ldots,\boldsymbol{x}_K)\|p(\boldsymbol{y}_1,\ldots,\boldsymbol{y}_K,\boldsymbol{z},c)) \tag{11}$$

with the set $\phi$ of all weight parameters in the classifier, encoder, and decoder deep nets. We compute the reconstruction term $\mathcal{L}_{\text{recon}}$ as follows:

$$\mathcal{L}_{\text{recon}} = \sum_{c=1}^{C}q(c|\boldsymbol{x}_1,\ldots,\boldsymbol{x}_K)\mathbb{E}_{q(\boldsymbol{y}_1,\ldots,\boldsymbol{y}_K,\boldsymbol{z}|\boldsymbol{x}_1,\ldots,\boldsymbol{x}_K,c)}\left[\sum_{k=1}^{K}\log p(\boldsymbol{x}_k|\boldsymbol{y}_k,\boldsymbol{z},c)\right] \tag{12}$$

$$\approx \sum_{c=1}^{C}q(c|\boldsymbol{x}_1,\ldots,\boldsymbol{x}_K)\sum_{k=1}^{K}\log p(\boldsymbol{x}_k|\boldsymbol{y}_k,\boldsymbol{z},c) \tag{13}$$

where we approximate the expectation using one sample $\boldsymbol{z} \sim q(\boldsymbol{z}|\boldsymbol{x}_1,\ldots,\boldsymbol{x}_K,c)$ and $\boldsymbol{y}_k \sim q(\boldsymbol{y}_k|\boldsymbol{x}_k,c)$ for each $k$, but directly use the probability value $q(c|\boldsymbol{x}_1,\ldots,\boldsymbol{x}_K)$ for $c$. The KL term $\mathcal{L}_{\text{KL}}$ is computed as follows:

$$\mathcal{L}_{\text{KL}} = -D_{\text{KL}}(q(c|\boldsymbol{x}_1,\ldots,\boldsymbol{x}_K)\|p(c))$$
$$-\sum_{c=1}^{C}q(c|\boldsymbol{x}_1,\ldots,\boldsymbol{x}_K)\sum_{k=1}^{K}D_{\text{KL}}(q(\boldsymbol{y}_k|\boldsymbol{x}_k,c)\|p(\boldsymbol{y}_k))$$
$$-\sum_{c=1}^{C}q(c|\boldsymbol{x}_1,\ldots,\boldsymbol{x}_K)D_{\text{KL}}(q(\boldsymbol{z}|\boldsymbol{x}_1,\ldots,\boldsymbol{x}_K,c)\|p(\boldsymbol{z})) \tag{14}$$

Finally, our training procedure is to maximize the lower bound for the entire dataset with respect to the weight parameters: $\hat{\phi} = \arg\max_{\phi}\frac{1}{N}\sum_{n=1}^{N}\mathcal{L}(\phi;\boldsymbol{x}_1^{(n)},\ldots,\boldsymbol{x}_K^{(n)})$. A diagrammatic outline of the algorithm is given in Figure 2.

## 4 EXPERIMENTS

We have applied the model described in Section 3 to two image datasets: ShapeNet (general objects) and MultiPie (natural faces). Below, we outline the experimental set-up and show the results.

### 4.1 SHAPENET

For the first set of experiment, we created a dataset of multi-viewed object images derived from 3D models in ShapeNet database (Chang et al., 2015). We selected 10, out of 55, pre-defined object classes that each have a relatively large number of object identities (car, chair, table, airplane, lamp,

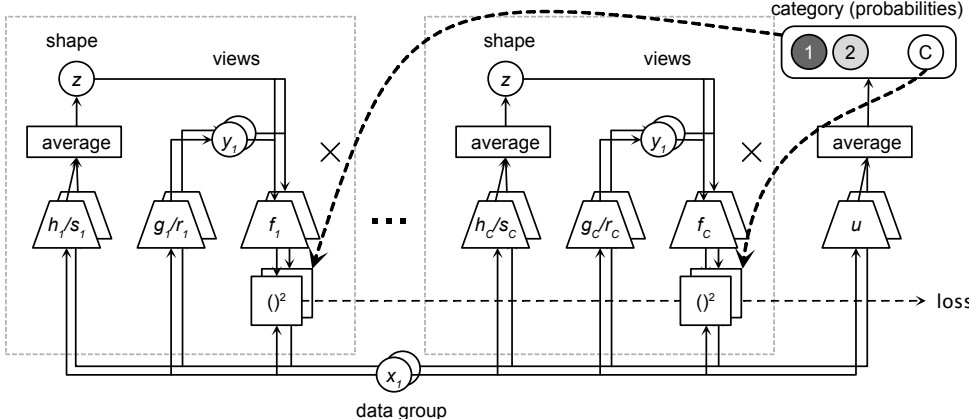

Figure 2: A diagrammatic outline of CIGMO learning algorithm. The structure of the entire work-flow consists of $C$ modules (light-gray boxes) of group-based VAE corresponding to $C$ categories and a classifier network (right-most). An input is a group of data instances $x_k$ (bottom). Given such input, each module $c$ estimates an instance-specific view $y_k$ from each data instance $x_k$ using the encoders $g_c/r_c$. The module also estimates a group-common shape $z$ by first estimating individual shapes for instances $x_k$ using the encoders $h_c/s_c$ and then by taking their average. Then, new data instances are generated by the decoder $f_c$ from the views and shape, and compared with the original input data for obtaining the reconstruction error (loss). This process is repeated for all modules. In parallel, the posterior probability for category $c$ is computed by the classifier $u$ on each data instance $x_k$ and then by averaging over the instances. Each computed probability is multiplied with the reconstruction error for the corresponding module. Other probabilistic mechanisms (e.g., priors) are omitted here for brevity.

boat, box, display, truck, and vase); we avoided heavily overlapping classes with many visually similar objects, e.g., chair, sofa, and bench. We then rendered each object in 30 views in a single lighting condition. We split the training and test sets, which consisted of 21888 and 6210 object identities, respectively. We also created subset versions with 2, 3, or 5 object classes. For training data, we formed groups of images of the same object in random 3 views ($K = 3$). We used object identity labels (not class labels) for forming grouped data, but, after this step, we never used any label during the training. We converted all images to gray-scale to make the model concentrate on shape information rather than color information, as the latter easily becomes obvious clues for clustering (Ji et al., 2019). See Appendix A for more detail on the dataset.

To train a CIGMO model, we used the following architecture. First, we set the number of categories in the model to the number of classes in the data ($C = 2, 3, 5$, or 10). We set the shape dimension $M = 100$ and the view dimension $L = 3$. Here, using a very low view dimension was crucial since otherwise the view variable $\boldsymbol{y}_k$ would take over all the information in the input and the shape variable $\boldsymbol{z}$ would become degenerate (Hosoya, 2019). The classifier deep net $u$ consisted of three convolutional layers and two fully connected layers and ended with a Softmax layer. The shape and view encoder deep nets had a similar architecture, except that the last layer was linear for mean encoding ($g_c$ and $h_c$) and ended with Softplus for variance encoding ($r_c$ and $s_c$). The decoder deep nets $f_c$ had an inverse architecture ending with Sigmoid. Since the model had so many deep nets, a large part of the networks was shared to save the memory space. See Appendix B for details of the architecture. For simplicity, we fixed the category prior $\pi_c = 1/C$. For optimization, we used Adam (Kingma & Ba, 2015) with mini-batches of size 100.

In addition to CIGMO models, we trained a number of related models for comparison. To investigate the effect of decoupled representation of category and shape, we trained GVAE models (Hosoya, 2019), which can be obtained as a special case of CIGMO with a single category ($C = 1$). To examine the effect of disentangling of shape and view, we trained mixture of VAEs, again obtained as CIGMO with no grouping ($K = 1$; the shape and view variables are integrated). We also trained plain VAEs for basic comparison ($C = K = 1$). For a part of evaluation below, since GVAE

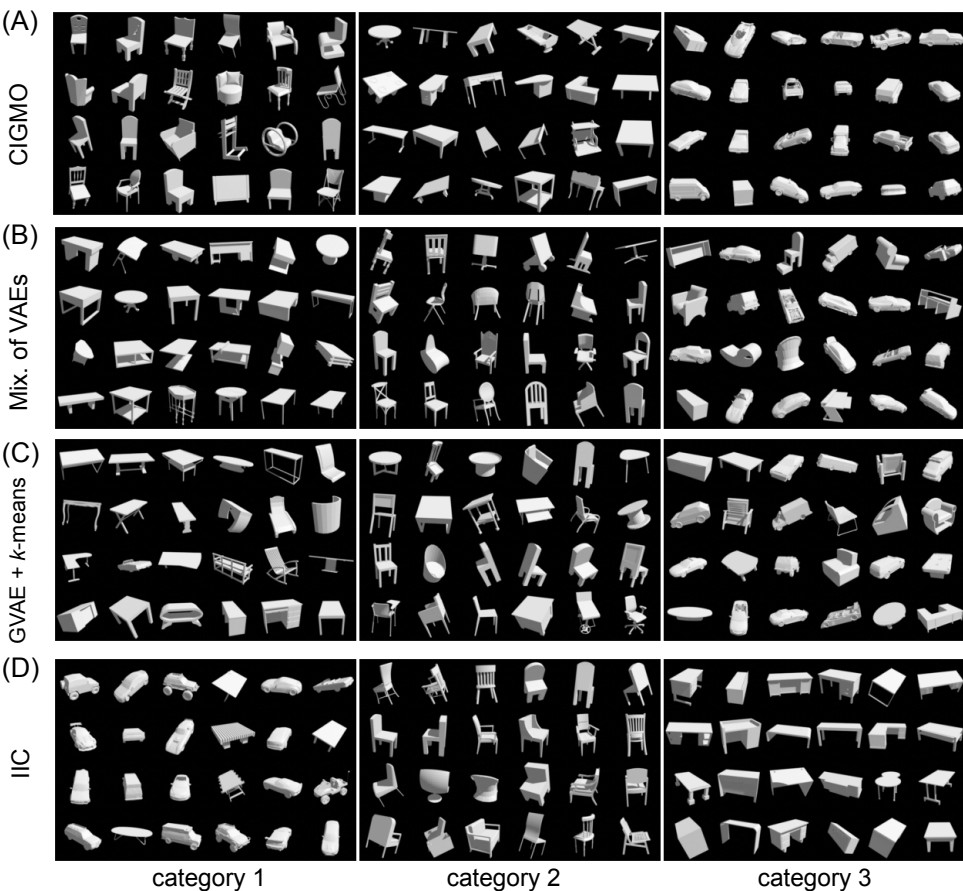

Figure 3: Examples of invariant clustering from (A) a CIGMO, (B) a mixture of VAEs, (C) a GVAE with $k$-means, and (D) an IIC, in the case of 3 categories. Random 24 images belonging to each estimated category are shown in a box. Note that the categories quite precisely correspond to the car, chair, and table image classes in (A), whereas such correspondence is less clear in the other models (B–D; in particular, cars are mixed with many other objects).

and VAE themselves have no clustering capability, we performed $k$-means on the shape variable of GVAE or the latent variable of VAE. In addition, we incorporated two completely different methods: Multi-Level VAE (MLVAE) (Bouchacourt et al., 2018), another group-based generative model, and Invariant Information Clustering (IIC) (Ji et al., 2019), a method specialized to invariant clustering. We tested two versions of IIC, with and without regularization using 5-times overclustering (Ji et al., 2019). For each method, we trained 10 model instances from the scratch.

We evaluated the trained models using test data, which were ungrouped and contained objects of the same classes as training data but of different identities. The evaluation involved three tasks: (1) invariant clustering, (2) one-shot classification, and (3) feature manipulation.

### 4.1.1 INVARIANT CLUSTERING

In this task, we simply inferred the most probable category from a given image, $\hat{c} = \arg\max_c q(c|\boldsymbol{x})$. Figure 3(A) illustrates results from a CIGMO model with 3 categories, where each box shows random images belonging to each estimated category. This demonstrates a very precise clustering of objects achieved by the model, which is quite remarkable given the large view variation and no category label used during training. Figure 3(B–D) shows analogous examples for alternative models (mixture of VAEs, GVAE with $k$-means, and IIC). The result shows a clear degrade of performance.

| # of categories | 2 | 3 | 5 | 10 |
|---|---|---|---|---|
| chance level | 50.00 | 33.33 | 20.00 | 10.00 |
| IIC | $95.12 \pm 5.70$ | $85.25 \pm 13.74$ | $81.10 \pm 7.33$ | $60.84 \pm 1.45$ |
| IIC (overcluster.) | $93.93 \pm 6.44$ | $79.86 \pm 13.78$ | $81.87 \pm 4.57$ | $59.73 \pm 1.49$ |
| VAE + $k$-means | $71.98 \pm 7.04$ | $66.41 \pm 5.69$ | $50.83 \pm 3.85$ | $37.07 \pm 1.00$ |
| Mix. of VAEs | $81.56 \pm 10.91$ | $82.35 \pm 5.66$ | $65.73 \pm 6.24$ | $40.86 \pm 3.58$ |
| MLVAE + $k$-means | $90.56 \pm 6.86$ | $82.04 \pm 7.78$ | $70.68 \pm 5.04$ | $54.47 \pm 1.92$ |
| GVAE + $k$-means | $90.07 \pm 4.97$ | $73.20 \pm 10.93$ | $69.42 \pm 3.47$ | $52.55 \pm 2.74$ |
| **CIGMO** | **98.41** $\pm$ 0.50 | **94.83** $\pm$ 6.06 | **89.36** $\pm$ 4.53 | **68.53** $\pm$ 4.24 |

Table 1: Invariant clustering accuracy (%). The mean and SD over 10 model instances are shown.

For comparison of the methods, we quantified the performance of invariant clustering in terms of classification accuracy. Here, we used the best category-to-class assignment computed by the Hungarian algorithm (Munkres, 1957). Table 1 summarizes the results. Generally, CIGMO outperformed the other methods in all cases with a large margin. More specifically, first, CIGMO always performed better than mixture of VAEs, showing the importance of shape-view disentangling. Second, CIGMO was also always better than GVAE (and MLVAE) plus $k$-means, also showing the importance of category-shape decoupling. In other words, if shapes of all categories were packed into a single latent variable, category information could not be clearly represented. These two points, taken together, emphasize the categorical and invariant nature of our model. Third, CIGMO gave performance superior to IIC, surpassing the state-of-the-art method for this task. This was the case both with and without overclustering; in fact, we could not find a consistent improvement by overclustering in IIC, contrary to the claim by Ji et al. (2019).

### 4.1.2 ONE-SHOT CLASSIFICATION

In this task, we split test data into gallery and probe, where gallery holds one image for each object and probe the rest, and then identify the object of each probe image. Note that our purpose here is not to infer the class but the object identity, unlike invariant clustering. Note also that, since the test objects are disjoint from the training objects, both gallery and probe images contain only unseen objects for the model. We used this task here since its performance can serve as a criterion for evaluating disentangled representations (Hosoya, 2019). The rationale is that, if shape code is perfectly invariant in view, then all images of the same object should be mapped to an identical point in the shape space.

Thus, we compared overall accuracy of one-shot classification for CIGMO and other models. For this, we performed a nearest-neighbor method according to cosine distance in the shape space. Here, the shape space was defined depending on the method. For GVAE (or MLVAE), the shape variable $z = h(x)$ directly defined the shape space. For CIGMO, since the shape representation depended on the category, we first constructed a $C \times M$ matrix $Z$ such that $Z_{c,*} = h_c(x)$ for $c = \hat{c}$ and $Z_{c,*} = 0$ otherwise, and then flattened the matrix to a vector. This gave us category-dependent shape vectors that could be directly compared and, in particular, those of different categories would give cosine distance 1, the maximum value. For VAE or mixture of VAEs, we used a similar scheme except that we used the entire latent variable in place of shape variable.

Table 2 summarizes the results. Overall, CIGMO performed the best among the compared methods in all cases. In particular, it outperformed, by far, mixture of VAEs, showing the success of shape information disentangled from view. Further, CIGMO also performed significantly better than GVAE and MLVAE, which indicates that shapes can be represented more efficiently with category specialization than without. These results, again, reveal the advantage of the categorical and invariant representations in modeling general object images. (Note also that the scores were remarkably high even for up to 6210-way classification by one shot.)

### 4.1.3 FEATURE MANIPULATION

CIGMO provides various ways of manipulating latent feature representations and generating new images. These include (1) swapping, to generate an image from the view of one image and the shape of another, (2) interpolation, to generate an image from the shape and view that linearly

| # of categories | 2 | 3 | 5 | 10 |
|---|---|---|---|---|
| chance level | 0.05 | 0.03 | 0.02 | 0.02 |
| VAE | $1.29 \pm 0.03$ | $1.81 \pm 0.04$ | $3.09 \pm 0.05$ | $2.97 \pm 0.02$ |
| Mix. of VAEs | $1.40 \pm 0.07$ | $1.91 \pm 0.06$ | $3.30 \pm 0.07$ | $3.15 \pm 0.06$ |
| MLVAE | $20.00 \pm 0.77$ | $20.56 \pm 0.50$ | $17.69 \pm 0.28$ | $15.68 \pm 0.28$ |
| GVAE | $19.71 \pm 0.87$ | $21.44 \pm 0.54$ | $17.85 \pm 0.28$ | $15.93 \pm 0.18$ |
| **CIGMO** | $\mathbf{22.10} \pm 0.99$ | $\mathbf{23.95} \pm 0.52$ | $\mathbf{21.66} \pm 0.79$ | $\mathbf{19.49} \pm 0.71$ |

Table 2: One-shot classification accuracy (%). The mean and SD over 10 model instances are shown.

| # of categories | 2 | 3 | 5 | 10 |
|---|---|---|---|---|
| MLVAE | $54.80 \pm 1.37$ | $53.59 \pm 0.57$ | $47.37 \pm 0.85$ | $42.99 \pm 0.63$ |
| GVAE | $55.35 \pm 1.35$ | $55.17 \pm 0.80$ | $48.23 \pm 0.58$ | $43.85 \pm 0.29$ |
| **CIGMO** | $55.33 \pm 1.47$ | $57.69 \pm 0.94$ | $51.00 \pm 0.89$ | $46.30 \pm 0.93$ |
| MLVAE | $0.21 \pm 0.02$ | $0.24 \pm 0.02$ | $0.62 \pm 0.04$ | $0.53 \pm 0.09$ |
| GVAE | $0.21 \pm 0.02$ | $0.23 \pm 0.04$ | $0.55 \pm 0.05$ | $0.48 \pm 0.05$ |
| **CIGMO** | $0.27 \pm 0.03$ | $0.26 \pm 0.04$ | $0.65 \pm 0.05$ | $0.66 \pm 0.08$ |

Table 3: Quality of shape-view disentanglement, measured as neural network classification accuracy (%) for object identity from the shape (top rows) or view variable (bottom rows); weighted average over categories. The mean and SD over 10 model instances are shown.

interpolates those of two images, and (3) random generation, to generate an image from shape and view randomly drawn from Gaussian distributions. Analogous manipulations have commonly been used in previous studies (Mathieu et al., 2016; Bouchacourt et al., 2018; Hosoya, 2019), but our cases are conditioned on a category. Appendix C gives examples of these feature manipulations, in which we can qualitatively confirm that our model performed these tasks as intended, e.g., reasonably clear alignment of views in rows and shapes in columns in swapping. Although generating sharp images was not the focus here, improvement in this direction could be done, e.g., by incorporating adversarial learning for regularization (Mathieu et al., 2016).

Relevant to these, we quantitatively evaluated the quality of shape-view disentanglement in each category. Specifically, we measured how much information the shape or view variable contained on object identity and, for this purpose, we trained two-layer neural networks on either variable for classification (Mathieu et al., 2016; Bouchacourt et al., 2018). A better disentangled representation was expected to give a higher accuracy from the shape variable and a lower accuracy from the view variable. We performed this for each category using the belonging test images and took the average accuracy weighted by the number of those images. Table 3 summarizes the results. Overall, CIGMO gives fairly comparable quality to the existing disentangling methods.

## 4.2 MULTIPIE

For the second set of experiment, we used a dataset of multi-viewed face images derived from MultiPie dataset (Gross et al., 2010). We followed the same data preparation as Hosoya (2019), except that we included the images in all lighting conditions and converted them to gray-scale. We split the training and test sets consisting of disjoint 795 and 126 identities, and grouped together the training images that have the same identity, hair/cloth style, and expression. For this dataset, there was no pre-defined class unlike ShapeNet, so we arbitrarily set the number of categories to 5 in CIGMO models. Otherwise, the training condition was the same as before.

Figure 4 illustrates the results from a CIGMO model on (A) invariant clustering and (B) category-wise swapping, where only 3 categories were shown as the other 2 categories were degenerate to which no input belonged. By inspection, the 3 effective categories seemed to represent faces with long hair, round faces with short hair, and oval faces with short hair, respectively, in a view-invariant manner (although this observation was not verified since the dataset lacked relevant labels). Other CIGMO model instances trained in the same way had 1 to 3 effective categories and seemed to have slightly different categorization strategies, but generally based on hair length, hair color, facial aspect ratio, beard, or skin color; we could not find categorization by expression. We also measured

(A) Invariant clustering

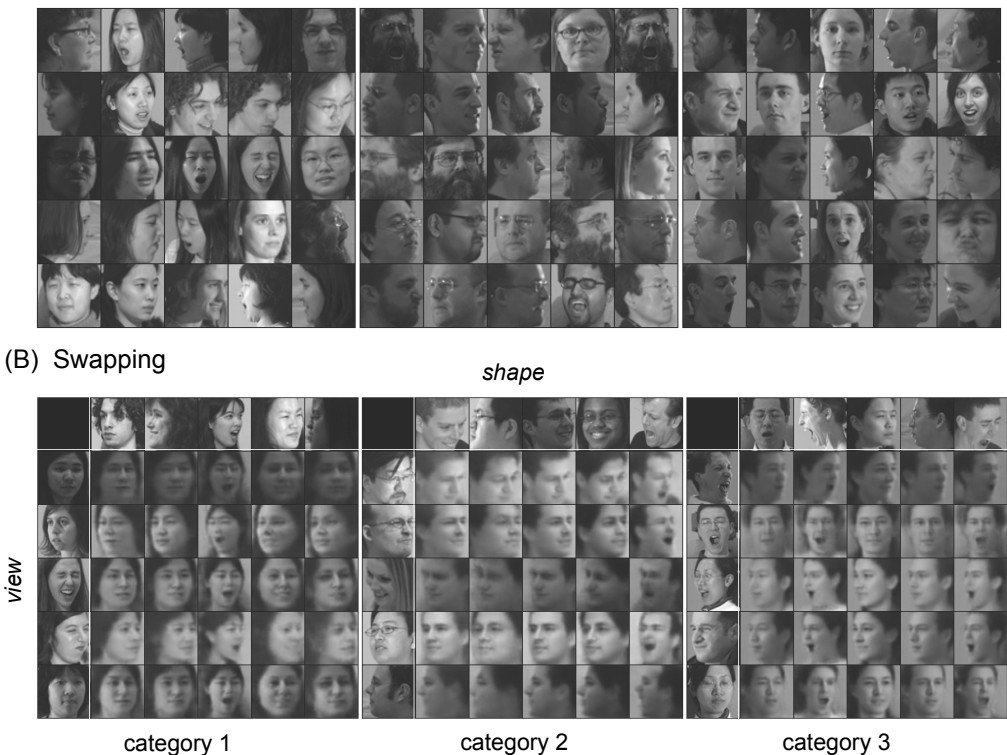

(B) Swapping

*shape*

*view*

category 1                    category 2                    category 3

Figure 4: Results from a CIGMO model trained for MultiPie dataset. (A) Invariant clustering. Random 25 images belonging to each estimated category are shown in a box. (B) Swapping. For each category, we generate a matrix of images from two lists of sample images, where each generated image has the view of an image in the left list and the shape of an image in the top list. Both together indicate that the model provides a shape-view disentangled representation of specific categories corresponding to faces with long hair, round faces with short hair, and oval faces with short hair.

one-shot classification accuracy and shape-view disentangling quality. However, CIGMO overall gave a similar performance to GVAE or MLVAE, indicating a lesser importance of category-shape decoupling in this task for this dataset. This is understandable since, in a sense, all faces look alike and therefore, in what way faces are categorized, cross-category diversity would not be so large compared to within-category diversity.

## 5    CONCLUSION

In this paper, we have proposed CIGMO as a deep generative model that can discover category, shape, and view factors from general object images. The model has the form of mixture of disentangling generative models and comes with a VAE-based algorithm that requires no explicit label but only view-grouping information. By application to two image datasets, we showed that our model can learn to represent categories in the mixture components and category-specific disentangled information of shape and view in each component model. We demonstrated that our model can outperform existing methods including state-of-the-art methods in invariant clustering and one-shot classification tasks, emphasizing the importance of categorical invariant representation. Future investigation will include improvement on image generation quality, category degeneracy, and scalability, and application to more realistic datasets. Lastly, CIGMO's biological relationship to the primate inferotemporal cortex would be interesting to pursue, as our present work was originally inspired so.

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

## A  DATASET DETAILS

Our object image dataset stemmed from a core subset of ShapeNet database of 3D object models (Chang et al., 2015) used in SHREC2016 challenge[1]. The subset contained 55 object classes and did not include material data. Out of the 55 classes, we selected 10 classes: car, chair, table, airplane, lamp, boat, box, display, truck, and vase. Our criterion here was to select classes with a large number of object identities but avoid including visually similar classes, e.g., chair, sofa, and bench. We rendered each object in 30 views consisting of 15 azimuths (equally dividing $360°$) and 2 elevations ($0°$ and $22.5°$ downward) in a single lighting condition; the images were gray-scale and had size $64 \times 64$ pixels. All the rendering used Blender software[2]. We divided the data into training and test following the split given in the original database. We also created versions of the dataset consisting of the first 2, 3, or 5 object classes listed above.

## B  ARCHITECTURE DETAILS

In a CIGMO model, the classifier deep net $u$ consisted of three convolutional layers each with 32, 64, and 128 filters (kernel $5 \times 5$; stride 2; padding 2), followed by two fully connected layers each with 500 intermediate units and $C$ output units. These layers were each intervened with Batch Normalization and ReLU nonlinearity, except that the last layer ended with Softmax. The shape and view encoder deep nets had a similar architecture, except that the last layer was linear for encoding the mean ($g_c$ and $h_c$) or ended with Softplus for encoding the variance ($r_c$ and $s_c$). The decoder deep nets $f_c$ had two fully connected layers (103 input units and 500 intermediate units) followed by three transposed convolutional layers each with 128, 64, and 32 filters (kernel $6 \times 6$; stride 2; padding 2). These layers were again intervened with Batch Normalization and ReLU nonlinearity, but the last layer was Sigmoid.

To save the memory space, the shape encoders shared the first four layers for all categories and for mean and variance. The view encoders shared the entire architecture for all categories, but with a separate last layer for mean or variance specification. The decoders shared all but the first layer for all categories.

In the quantitative comparison, we obtained a mixture of VAEs, a GVAE model, and a VAE model as a special case of CIGMO model. Namely, a mixture of VAEs was a CIGMO with no grouping ($K = 1$), a GVAE was a CIGMO with a single category ($C = 1$), and a VAE was a CIGMO with both constraints ($K = 1$ and $C = 1$). In the case of no grouping, since no structure could differentiate the shape and view dimensions, we combined these into a single latent variable of 103 dimensions.

## C  ADDITIONAL RESULTS

We performed the following feature manipulation tasks introduced in Section 4.1.3.

---

[1] https://shapenet.cs.stanford.edu/shrec16/
[2] https://www.blender.org

**Swapping** For a category $c$ and for images $x_1$ and $x_2$ belonging to $c$, obtain $x_{\text{swap}} = f_c(y_1, z_2)$ where $y_1 = g_c(x_1)$ and $z_2 = h_c(x_2)$.

**Interpolation** For a category $c$ and for images $x_1$ and $x_2$ belonging to $c$, obtain $x_{\text{interp}} = f_c(\alpha y_1 + (1-\alpha)y_2, \beta z_1 + (1-\beta)z_2)$ where $y_i = g_c(x_i)$ and $z_i = h_c(x_i)$ with $0 \le \alpha, \beta \le 1$ and $i = 1, 2$.

**Random generation** For a category $c$, obtain $x_{\text{rand}} = f_c(y, z)$ with $y \sim \mathcal{N}_L(0, I)$ and $z \sim \mathcal{N}_M(0, I)$.

Figure 5 shows examples of these feature manipulations on a 3-category CIGMO model trained on ShapeNet. As we can see, swapping gives a clear alignment of views in rows and shapes in columns. Interpolation gives smooth changes of images from one image to another in both shape and view dimensions. Random generation gives new images most of which are recognizable as each category.

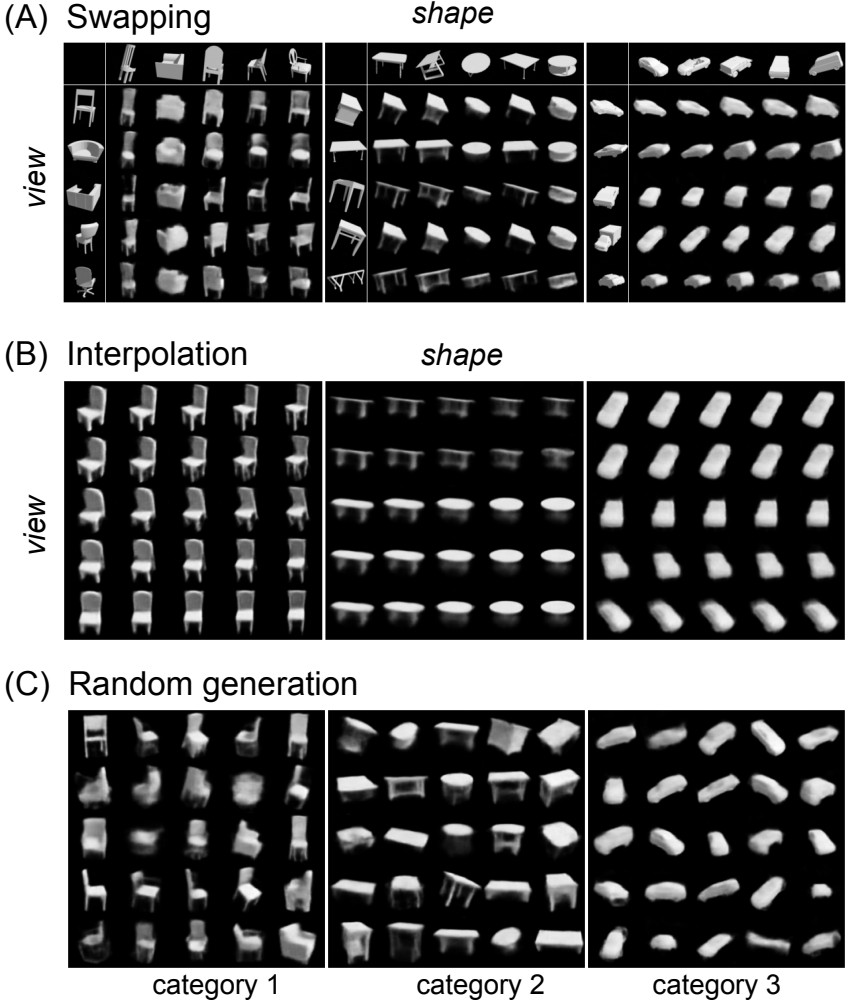

Figure 5: Examples of feature manipulation tasks from a 3-category CIGMO model for ShapeNet dataset. (A) Swapping. For each category, we generate a matrix of images from two lists of sample images, where each generated image has the view of an image in the left list and the shape of an image in the top list. (B) Interpolation. For each category, we generate a matrix of images from two sample images (corresponding to the top-left and bottom-right images), where each generated image has the view and shape that linearly interpolates those of the two images. (C) Random generation. For each category, we generate images from shapes and views that are drawn from Gaussian distributions.

