# OpenReview forum: "CIGMO: Learning categorical invariant deep generative models from grouped data"
_ICLR.cc/2021/Conference — Reject_

### Official Review · AnonReviewer3 · 2020-10-15
**Incremental improvement over GVAE**

**Rating:** 4
**Confidence:** 4

**Review:**

This work proposes a probabilistic model which disentangles view, class and shape attributes explicitly (it does not rely on an emergent phenomena of disentangled factors in the latent space). In comparison to a similar approach, GVAE, CIGMO additionally disentangles the content factor into category and shape factors. Obtained results are reasonable and show advantage of the proposed method over related approaches.

Strengths:
- I believe the idea, while not being novel compared to GVAE, shows that one can scale VAEs into disentangling more factors. I imagine the model that can additionally incorporate the data where additional factors are present such as explicit camera viewpoint.
- Theoretical grounding is well formulated.
- Baselines are well chosen and show superiority over the basic baseline that came to my mind while reading this work - mixture of VAEs.


Weaknesses:
- The authors mentioned that their approach is heavily influenced by GVAE but do not mention how these two differ in any principal way. I perceive CIGMO as an incremental improvement over GVAE. To mitigate this issue, it would be great to see this work as a meta approach over GVAE where the authors include more factors that are easily obtainable (since ShapeNet is rendered either way, hence the authors can extract illumination, camera position and other factors) and how including these factors influence the results. With the presented model formulation, it seems to be possible to include more such factors. Finally, I would argue that such a meta approach would increase the applicability and extensibility of the work significantly.
- Choosing 10 classes is not well supported. What classes are chosen? Does the final dataset include classes that are rare in the ShapeNet (besides, it would show how the model is able to handle such objects thanks to the new class factor)
- The main part lacks a good diagram of the method. The plate model is not sufficient since details that make the model converge in the first place cannot be presented in such a diagram. The one presented in the Appendix does more harm than good and I spent more time to understand it than to follow equations directly.

Other comments:
- Overall, the paper is well written, however the authors should focus on typos present in the paper. Also, fixing or reformulating the sentence “our choice [ … ] is orthogonal to our goal” may reveal the main difference with GVAE.
The authors refer to biological motivation which is not mentioned later in the work. In this case, the whole paragraph is highly irrelevant to me. It is also formulated in such a way that one would argue that mammals are limited in terms of how many objects they can recognize by shape.

---

> ### Author Response · Authors · 2020-11-18
> **Response to AnonReviewer3**
>
> Q1: The authors mentioned that their approach is heavily influenced by GVAE but do not mention how these two differ in any principal way. I perceive CIGMO as an incremental improvement over GVAE.
>
> The factors that GVAE can separate are fundamentally two---a group-common factor (content) and an instance-specific factor (view)---and there is no simple way to extend this to more than two.  CIGMO does this by introducing a more complicated machinery, that is, we define a mixture model where each component model is a GVAE (with shape and view variables) and then fit the mixture model to a grouped dataset. This gives rise to the category factor, and this happens even if the grouping information is not related to any category information.  We believe that this leap from GVAE to CIGMO is indeed fundamental and significant.  In the revision, we emphasize this point (Sec 2).
>
> Q2: Choosing 10 classes is not well supported. What classes are chosen? Does the final dataset include classes that are rare in the ShapeNet (besides, it would show how the model is able to handle such objects thanks to the new class factor).
>
> We basically selected the 10 classes so that each class had a sufficiently large number of objects.  This is because some classes contained only a few tens of objects, which was not enough to learn a view-invariant representation.  We also avoided including overly ambiguous classes, e.g., chair, bench, and sofa, which overlap badly with each other (indeed some visually similar images were labeled chair or bench).  We had this description in the Appendix, but move it to the main part in the revision (Sec 4.1).
>
> Q3: The main part lacks a good diagram of the method. The plate model is not sufficient since details that make the model converge in the first place cannot be presented in such a diagram. The one presented in the Appendix does more harm than good and I spent more time to understand it than to follow equations directly.
>
> In the revision, we put the diagram in Appendix to the main part (Fig 2) and improved the explanation quite a bit.  We hope that these are clearly enough.
>
> Q4: The authors should focus on typos present in the paper.
>
> In the revision, we worked through the paper to fix typos.
>
> Q5: Also, fixing or reformulating the sentence “our choice [ … ] is orthogonal to our goal” may reveal the main difference with GVAE.
>
> In the revision, we clarify the difference from GVAE (Sec 2).
>
> Q5: The authors refer to biological motivation which is not mentioned later in the work. In this case, the whole paragraph is highly irrelevant to me. It is also formulated in such a way that one would argue that mammals are limited in terms of how many objects they can recognize by shape.
>
> We do not claim in this study that CIGMO can explain neural activities in the higher visual cortex.  However, we prefer to keep this statement since it was indeed our inspiration.  In the revision, we mention future possibility to relate CIGMO to biology (Sec 5).

---

### Official Review · AnonReviewer2 · 2020-10-19
**Interesting paper with some novelty but the experimental results are lacking to clearly demonstrate the merits.**

**Rating:** 5
**Confidence:** 4

**Review:**

Strength:
- The proposed model can simultaneously model the category, shape, and view factors of general object images.
- The supervision is only from the view-grouping information, which can be easily obtained.

Weakness:
- The proposed model is a mixture of disentangling models, which is not in an end-to-end manner.
- The authors only compare the proposed model with other methods in two down-stream tasks. The authors should compare the generated results directly in quantitative and qualitative ways.
- Experimental results are not convincing.  The quantitative comparisons of invariant clustering and one-shot classification are interesting settings and clearly show good performance of the proposed method. However, it does not seem very fair to compare methods with differently structured input. As the GVAE baseline does not model category (C=1) and the mixture of VAEs are trained with ungrouped data (K=1), it is easy to see why the proposed method performs better in such tasks. Adding a stronger baseline with the same level of supervision (grouped input) would make the comparison more convincing. In Figure 3 (B), the qualitative results of different views are not clearly shape-invariant. It would be clearer if the authors can show the swapping results on the ShapeNet objects (and possibly of other baseline methods).

While the authors have validated the proposed model on the ShapeNet and MutliPIE datasets, the variations in these images are limited with simple backgrounds (in well-controlled lab setups). It is important to demonstrate whether the proposed method performs well on real-world datasets (e.g., LFW face, Pascal3D+).

Novelty: The novelty of this work is generally good. The proposed model can discover the category, shape, and view factor at once, so as to generate general object images in a much larger space. The model capacity is much stronger than previous generative models which try to model general object images. What’s more, the model is trained in a weakly-supervised manner. The supervision information can be easily obtained.

Presentation: The mathematical representation is easy to understand theoretically. On the other hand, the authors should give more initiative representations when describing the workflow of the proposed model and training procedure, which can make the paper much easier to understand. In addition, there should be more implementation details in the paper.

Evaluation: As discussed above, the authors only compare their method with other methods in two downstream tasks. As such, the experimental results are not convincing. Could the authors directly show the comparison results between the proposed method and other approaches quantitively and qualitatively?  For example, as the authors claimed, the proposed method can model the category, shape, and viewpoint of images at once, but other previous models cannot achieve this goal. Can the authors directly compare their results with the results from other models to support this claim?  The authors should provide such results to support their claim more directly.

Contribution: The contributions of this paper are good, which can simultaneously discover those factors of general object images.
Nevertheless, the experimental results are lacking to clearly demonstrate the merits of this work.

---

> ### Author Response · Authors · 2020-11-18
> **Response to AnonReviewer2**
>
> Q1: The proposed model is a mixture of disentangling models, which is not in an end-to-end manner.
>
> Actually, our training scheme is end-to-end once a grouped data is given.  It is certainly not end-to-end in terms of a bare dataset without grouping information.  However, we believe that, in that case, it is fundamentally impossible to disentangle category, shape, and view without any inductive bias.  In the revision, we have related discussion (Sec 2).
>
> Q2: The authors only compare the proposed model with other methods in two down-stream tasks. The authors should compare the generated results directly in quantitative and qualitative ways.
>
> Please note first that the invariant clustering task measures how well the estimated categories match with the ground-truth class labels, which should be a direct evaluation in terms of categories.  However, although one-shot classification performance would somewhat measure the quality of disentanglement between shape and view, this confounded clustering accuracy and shape-view disentanglement and therefore may not be a direct measure for shape-view disentanglement.
>
> Therefore, in the revision, we added another, more direct, quantitative evaluation: category-averaged accuracy classification of shape variables or view variables for object identities using two-layer neural networks.  A good disentangling model would give a high accuracy for shape variables and a low accuracy for view variables.  This is an adaption of a standard approach used very commonly in the literature on group-based disentangling models.  The result shows that the quality of shape-view disentanglement was comparable to the compared models (Table 3).
>
> Q3: The performance evaluation does not seem very fair to compare methods with differently structured input. As the GVAE baseline does not model category (C=1) and the mixture of VAEs are trained with ungrouped data (K=1), it is easy to see why the proposed method performs better in such tasks.  Adding a stronger baseline with the same level of supervision (grouped input) would make the comparison more convincing.
>
> Indeed, comparison with GVAE and mixture of VAEs was a kind of empirical check of our conception of CIGMO.  However, please note that, for the invariant clustering task, we actually did give comparison with IIC, which is a completely different method that works on grouped data.  In addition, in the revision, we show additional results comparing with another different group-based model, MLVAE, for both invariant clustering and one-shot classification (Table 1,2).  In all of these, CIGMO is superior to the other methods.
>
> Q4: In Figure 3 (B), the qualitative results of different views are not clearly shape-invariant. It would be clearer if the authors can show the swapping results on the ShapeNet objects (and possibly of other baseline methods).
>
> In the submission, the generated images for MultiPIE were rather low-quality since the size of our training data was rather small.  That is, in our setting with only one lighting condition, we had only 2136 training images per view and, in the case of 3 categories, only one third of the 2136 images per view, on average, were effectively used for training each category.
>
> We therefore redid the entire model training on MultiPIE with a larger training set including all lighting conditions (previously a single condition).  As a result, the quality of image generation was quite a bit improved, while overall results were otherwise similar.  In particular, the identities of the generated face images are more easily distinguishable.  In the revision, we replace all relevant figures and descriptions accordingly (Fig 4).  (We keep the placement of the swapping matrix for ShapeNet in the Appendix since we think the image quality issue was resolved.)
>
> Q5: It is important to demonstrate whether the proposed method performs well on real-world datasets (e.g., LFW face, Pascal3D+).
>
> We agree that this direction is important.  However, it raises an additional challenge due to  the lower quality of the datasets, and therefore is left for future work.  We note this future possibility (Sec 5).
>
> Q6: The authors should give more initiative representations when describing the workflow of the proposed model and training procedure, which can make the paper much easier to understand. In addition, there should be more implementation details in the paper.
>
> In the revision, we moved the diagram in Appendix to the main part (Fig 2) and added more details in the caption.  We describe implementation details such as data preparation and architectures in Appendix.  Since these do not affect the overall results and we run out of space, we keep these in Appendix and hope that this is OK.

---

### Official Review · AnonReviewer4 · 2020-10-27
**Official Blind Review #4**

**Rating:** 7
**Confidence:** 3

**Review:**

============================
Summary

This paper proposes a categorical invariant generative model (CIGMO) from a set of 2D images that tries to disentangle the factors of data category, intra-category geometry, and rendering viewpoint. CIGMO trains a VAE with a hierarchical graphical model that explicitly factors out the three components by design. Experiments on two datasets (ShapeNet rendered images, MultiPie face images) show that the proposed method can discover the concept of data category without using explicit supervision. It also supports feature manipulation over the geometry and viewpoint factors.


============================
Pros

1) The proposed task and approach are valid, reasonable and technically sound. Also, the paper is very easy to follow.
2) This paper brings in a novel approach that explicitly models data category as a crucial factor for learning disentangled representations.
3) Experiments show that given no supervision, the clustering based on the category latent variables makes sense. And, with 1-shot supervision, the classification performance is better than baseline methods.
4) The learned geometry and viewpoint factors are well disentangled, as shown in the manipulation tasks.


============================
Cons

1) Can you show some quantitative evaluations or quantitative user studies to show the quality of the learned geometry and viewpoint disentanglement? How would you compare to previous works? I understand that your primary goal is to introduce data category factor into the game, but it is helpful for readers to know how much your model perform in terms of disentangling the standard two factors: geometry and viewpoint, and how do you compare to previous works?
2) In the manipulation experimental figures you show in the paper, the reconstruction quality seems pretty bad. Especially for the experiments on face dataset (Fig 3, B), it's really hard for me to tell the person identity from the reconstructed face images. Can you provide more discussions/explanations on this?

============================
Overall Rating

I like the novel task/method in this paper that brings in explicit modeling for the object category factor. Experiments comparing to baseline methods on unsupervised object category clustering and 1-shot object classification is quite convincing to me.

---

> ### Author Response · Authors · 2020-11-18
> **Response to AnonReviewer4**
>
> Q1: Can you show some quantitative evaluations or quantitative user studies to show the quality of the learned geometry and viewpoint disentanglement? How would you compare to previous works?
>
> It was our intention that one-shot classification performance would measure the quality of disentanglement between shape (geometry) and view.  However, since this measure confounded clustering accuracy and shape-view disentanglement, it may not be a direct measure for shape-view disentanglement.
>
> Therefore, in the revision, we added another, more direct, quantitative evaluation: category-averaged accuracy classification of shape variables or view variables for object identities using two-layer neural networks.  A good disentangling model would give a high accuracy for shape variables and a low accuracy for view variables.  This is an adaption of a standard approach used very commonly in the literature on group-based disentangling models.  The result shows that the quality of shape-view disentanglement was comparable to the compared models (Table 3).
>
> Q2: In the manipulation experimental figures you show in the paper, the reconstruction quality seems pretty bad. Especially for the experiments on face dataset (Fig 3, B), it's really hard for me to tell the person identity from the reconstructed face images. Can you provide more discussions/explanations on this?
>
> In the submission, the generated images for MultiPIE were rather low-quality since the size of our training data was rather small.  That is, in our setting with only one lighting condition, we had only 2136 training images per view and, in the case of 3 categories, only one third of the 2136 images per view, on average, were effectively used for training each category.
>
> We therefore redid the entire model training on MultiPIE with a larger training set including all lighting conditions (previously a single condition).  As a result, the quality of image generation was quite a bit improved, while overall results were otherwise similar.  In particular, the identities of the generated face images are more easily distinguishable.  In the revision, we replace all relevant figures and descriptions accordingly (Fig 4).

---

### Official Review · AnonReviewer1 · 2020-10-28
**This paper proposed a generative method, which learns to disentangle the category, shape, and view features of an image using weak supervision. The weak supervision is provided by grouping the images of same shape together and enforcing them to converge in the feature space. The authors performed two tasks: invariant clustering and one-shot classification, to demonstrate the effectiveness of the proposed method.**

**Rating:** 4
**Confidence:** 3

**Review:**

Strong Points:
1) The paper was fairly well written and easy to follow.
2) The results are quite promising.

Questions to the author:
1) It is unclear how the model learns the category without any supervision. It is recommended to provide more explanations as it is the major contribution.
2) It is unclear how \pi_c in equation 2 is determined. Does it follow the same distribution of the dataset?
3) For figure 2, why was the proposed CIGMO not compared to GVAE plus k-means or IIC?
4) For figure 3A, are the numbers of examples balanced in the 3 categories?
5) For figure 3B (swapping), based on data preparation, shape should capture identity, but it is not obvious from the figure. Any thoughts?
6) For the experiment with MultiPIE, grouping by emotion category might reveal more about the model.
7) The model is C(category) times larger than GVAE. Is it possible that the performance gain comes from using larger model?

---

> ### Author Response · Authors · 2020-11-18
> **Response to AnonReviewer1**
>
> Q1: It is unclear how the model learns the category without any supervision.
>
> The idea is that we first define a mixture model where each component model describes grouped data via shape and view variables and then we fit this parametric model to a dataset that groups together multiples views of the same object, which results in emergence of object categories as mixture components.  We note this in the revision (Sec 3.1).
>
> Q2: It is unclear how \pi_c in equation 2 is determined. Does it follow the same distribution of the dataset?
>
> In our experiment, for simplicity, \pi_c is fixed to 1/C (uniform).  We note this in the revision (Sec 4.1).  (We actually tried a version that learns \pi_c but the results were more or less the same.)
>
> Q3: For figure 2, why was the proposed CIGMO not compared to GVAE plus k-means or IIC?
>
> We show both results in the revision (Fig 3).
>
> Q4: For figure 3A, are the numbers of examples balanced in the 3 categories?
>
> This particular model instance was highly unbalanced: 1374 / 120 / 4176 test images belonged to each category.  This reflects the fact that the dataset contains only a small number of faces with light-colored hairs.  However, in the revision, we actually redid the whole training on MultiPIE, so replaced this figure.  See Q5.
>
> Q5: For figure 3B (swapping), based on data preparation, shape should capture identity, but it is not obvious from the figure. Any thoughts?
>
> In the submission, the generated images for MultiPIE were rather low-quality since the size of our training data was rather small.  That is, in our setting with only one lighting condition, we had only 2136 training images per view and, in the case of 3 categories, only one third of the 2136 images per view, on average, were effectively used for training each category.
>
> We therefore redid the entire model training on MultiPIE with a larger training set including all lighting conditions (previously a single condition).  As a result, the quality of image generation was quite a bit improved, while overall results were otherwise similar.  In particular, the identities of the generated face images are more easily distinguishable.  In the revision, we replace all relevant figures and descriptions accordingly (Fig 4).
>
> Q6: For the experiment with MultiPIE, grouping by emotion category might reveal more about the model.
>
> In the experiment, we actually grouped together images that have the same identity and emotion.  Nevertheless, the model learned, as categories, not emotion types but hair lengths, etc., probably because these were visually more prominent.  In the revision, we note this point (Sec 4.2).
>
> Q7: The model is C(category) times larger than GVAE. Is it possible that the performance gain comes from using larger model?
>
> This is unlikely since our model shares a large part of the encoder and decoder deep nets among categories: a 10-category CIGMO had 14.7M parameters in total while a GVAE had 13.3M.

---

### Author Response · Authors · 2020-11-18
**Notes on the first revision**

[This note has been updated to be more informative.]

We thank very much the reviewers for valuable comments and questions.  We have revised the manuscript based on these and this made a significant improvement in solidity and clarity.

The major changes in the revision are as follows.
1. One reviewer commented that the baseline models compared were only special cases of CIGMO.  Although we actually had a non-special case model (IIC) in the invariant clustering task, we indeed did not have such models in the one-shot classification task.  Therefore, we added another non-special-case baseline model, MLVAE, for all tasks (Table 1,2,3).  The result re-confirms the advantage of CIGMO over existing methods.
2. Two reviewers commented that we did not quantitatively evaluate shape-view disentanglement.  Although our initial intention was that the one-shot classification would serve for this, we agree that it somewhat confounded with invariant clustering so that the comparison was not very direct.  Therefore, we added another quantitative evaluation to more directly measure the quality of shape-view disentanglement (Table 3).  The result shows that the shape-view-disentanglement is comparable to existing methods.
3.  Three reviewers pointed out that the image generation from the models on MultiPIE was not good so that shape invariance was not clearly discernible.  After some analysis, we realized that this was because the dataset size used for this experiment was not large enough for obtaining a reasonable quality of image generation.  Therefore, we redid the entire experiment on MultiPIE using a much larger dataset (including all lighting conditions rather than a single condition).  As a result, the generated images look much better so that the identities are more distinguishable.

Below list the specific modified parts in the manuscript.  More details are given in the response to each reviewer.

Figure 1: The baseline graphical models (D,E,F) are removed to save space.

Sec 2: The first two paragraphs are rewritten so that the difference of CIGMO from existing group-based methods is emphasized.

Sec 3.1: The second half is rewritten for clarity.

Figure 2: The figure is moved from Appendix with a few edits (such as light-gray boxes).  The caption is largely rewritten for clarity.

Sec 4.1: The first paragraph is rewritten for clarifying the data preparation and for additional experiment on MLVAE.

Figure 3: Clustering results are newly added for (C) GVAE + kmeans and (D) IIC.

Table 1, 2: The results for MLVAE are added.

Table 3: New results on shape-view disentanglement are added.

Sec 4.1.3: The second paragraph is newly added in relation to the additional disentangling evaluation in Table 3.

Sec 4.2: The section is rewritten to incorporate the new experiment on MultiPIE.

Sec 5: Some sentences are added for future work.

Figure 4: The figure is replaced with a new one reflecting the new experiment on MultiPIE.

---

### Decision · Program_Chairs · 2021-01-07
**Final Decision**

**Decision:**

Reject

**Comment:**

Description:
The paper presents a weakly-supervised model CICGMO for disentangling category, shape and view information from images. Label information is not need as  the weak supervision is done by grouping together different views of the same object. They show that this outperforms other techniques on tasks such as invariant clustering and one-shot classification.

Strengths:
- Paper is well written
- Data category is explicitly modeled
- The weakly supervision approach is appealing,  since the grouped data used as supervision information is easy to obtain
- Invariant clustering and one-shot classification results outperforms other methods significantly, showing CIGMO is doing a decent job at those tasks. This could be explained by CIGMO ability to better  disentangle category-shape-view.

Weaknesses:
- It is unclear how well (quality) the generative model is able to disentangle shape from view
- The reconstruction quality is quite low, such that it is difficult often times, in the MULTIPIE example, to clearly identify a face geometry.
- Generated results are not evaluated directly, but rather evaluation is done through down-stream tasks such as invariant clustering. This makes it difficult to show the quality of shape and view information.